# Transfer learning with convolutional neural networks for cancer survival prediction using gene-expression data

**Guillermo López-García** *, **José M. Jerez, Leonardo Franco, Francisco J. Veredas**

Departamento de Lenguajes y Ciencias de la Computación, Universidad de Málaga, ETSI Informática, Málaga, Spain

* guilopgar@uma.es

## Abstract

Precision medicine in oncology aims at obtaining data from heterogeneous sources to have a precise estimation of a given patient's state and prognosis. With the purpose of advancing to personalized medicine framework, accurate diagnoses allow prescription of more effective treatments adapted to the specificities of each individual case. In the last years, next-generation sequencing has impelled cancer research by providing physicians with an overwhelming amount of gene-expression data from RNA-seq high-throughput platforms. In this scenario, data mining and machine learning techniques have widely contribute to gene-expression data analysis by supplying computational models to supporting decision-making on real-world data. Nevertheless, existing public gene-expression databases are characterized by the unfavorable imbalance between the huge number of genes (in the order of tenths of thousands) and the small number of samples (in the order of a few hundreds) available. Despite diverse feature selection and extraction strategies have been traditionally applied to surpass derived over-fitting issues, the efficacy of standard machine learning pipelines is far from being satisfactory for the prediction of relevant clinical outcomes like follow-up end-points or patient's survival. Using the public Pan-Cancer dataset, in this study we pre-train convolutional neural network architectures for survival prediction on a subset composed of thousands of gene-expression samples from thirty-one tumor types. The resulting architectures are subsequently fine-tuned to predict lung cancer progression-free interval. The application of convolutional networks to gene-expression data has many limitations, derived from the unstructured nature of these data. In this work we propose a methodology to rearrange RNA-seq data by transforming RNA-seq samples into gene-expression images, from which convolutional networks can extract high-level features. As an additional objective, we investigate whether leveraging the information extracted from other tumor-type samples contributes to the extraction of high-level features that improve lung cancer progression prediction, compared to other machine learning approaches.

**Data Availability Statement:** The data underlying the results presented in this study are publicly accessible from the UCSC Xena data browser. Concretely, two public datasets have been used:

the TCGA Pan-Cancer gene-expression dataset, and the TCGA Pan-Cancer curated clinical dataset. In addition, we have organized all the code and data needed to reproduce the results obtained in this paper in a publicly accessible repository: https://github.com/guilopgar/GeneExpImgTL.

**Funding:** This work was supported by the project TIN2017-88728-C2-1-R, MINECO, Plan Nacional de I+D+I, and by the Grupo de Investigación TIC226, Plan Andaluz de Investigación, Desarrollo e Innovación (PAIDI), Junta de Andalucía. The funders had no role in study design, data collection and analysis, decision to publish, or preparation of the manuscript.

**Competing interests:** The authors have declared that no competing interests exist.

## Introduction

In recent years, the analysis of gene-expression data is gaining a growing interest in the area of precision medicine, which establishes a framework for the advance and research in a series of clinical procedures aimed at making medicine more participatory, personalized, preventive and predictive (adjectives from which the concept of "P4 medicine" is also derived). In this context, the emergence of Next Generation Sequencing (NGS) techniques has allowed the transformation of areas such as precision medicine, biology or biochemistry. An unprecedented amount of data has been generated from which disciplines such as genomics, proteomics, transcriptomics, epigenetics or metabolomics are favored [1]. In particular, in clinical areas the appearance of high-throughput sequencing technology called RNA-Seq [2] has provided physicians with gene-expression data that allow more precise diagnosis and determination of the patient's state from a molecular point of view. Since cancer is a heterogeneous disease driven by diverse genomic alterations [3], the analysis of gene-expression data obtained from tumor tissue samples allows the study of molecular factors contributing to disease progression over time or influencing patient's survival. Gene-expression data contain valuable information on the levels of differential activation of the genes involved in the development and evolution of cancer. If that information is extracted effectively, it can leverage precise diagnostic methods leading to greater efficacy of treatments and better prognosis.

Although high-throughput sequencing techniques can supply oncologists and researchers with hundreds of thousands of gene-expression features, only a few of them are usually considered as clinically significant and subsequently utilized in practice for medical diagnosis and prognosis prediction [4, 5]. Accurately solving cancer prediction tasks—such as sub-type classification, survival prediction or time to progression estimation—from high-dimensional gene-expression data remains an open challenge for personalized genomic medicine. Training common survival and progression models, such as Cox proportional hazards, on high-dimensional gene-expression data would require a huge number of samples, which is a real handicap for this kind of traditional approaches. Thus, the number ($M$) of available gene-expression samples (i.e. the size of available cohorts) is usually much smaller (300-1$K$) than the number ($N$) of genes (10$K$-60$K$), which is known as the "curse of dimensionality" [6]. In this scenario, traditional survival analysis ends up selecting a reduced number or characteristics, thus the resulting models are usually prone to bias.

In the last decades, many machine learning (ML) approaches have been adapted to deal with cancer diagnosis and prediction on the basis of gene-expression data [7, 8]. On the one hand, prior knowledge has been used to select specific genes conforming gene signatures (also called biomarkers) that can be used to successfully predict certain clinical outcomes such as recurrence, treatment benefits, metastasis, etc. [9, 10]. On the other hand, different automatic feature extraction and selection approaches have been also used with the aim of reducing the dimensionality of the input space, as a previous step to applying different ML models [11]. Some methods, such as elastic net, have been adapted to regularize Cox models, aiming at reducing the number of input characteristics to solve survival prediction tasks [12]. Survival predictive models have been also addressed with decision-tree techniques, which have demonstrated to be resistant to over-fitting issues in high-dimensional scenarios [13]. Finally, neural networks (NN) approaches have been widely utilized in survival-prediction tasks on low-dimensional data [14], but the effectiveness of these methods for right-censored data does not seem to outperform that of Cox regression models as well it is strongly dependent on the underlying data structure [15]. In addition, the "black-box" nature of NNs makes them difficult to be interpretable in order to extract biological or clinical knowledge from the resulting models.

Deep learning (DL) encompasses a set of techniques—such as multi-layer neural networks (MLNN), convolutional neural networks (CNN), deep auto-encoders (AE) or recurrent neural networks (RNN)—that have recently achieved state-of-art performance in computer vision, pattern recognition and natural language processing [16]. In the particular case of high-throughput genomics, DL has demonstrated to be able to capture the internal structure of biological data and extract high-level abstract features from high-dimensional sequencing or expression data [17], thus improving performance and interpretability of traditional ML methods. In this way, recent studies have applied deep stacked or variational AEs to gene-expression public datasets for cancer prediction, some of them using transfer-learning (TL) approaches [18–21]. AEs have been also successfully applied on gene-expression or multi-omics data to tackle more complex and biologically-relevant cancer prediction tasks, such as classifying cancer sub-types or characterizing functional gene profiles [22–24]. Feed-forward MLNNs have been recently utilized to predict clinical outcomes from high-dimensional genomic data [25] or to identify cancer sub-types by combining supervised and unsupervised learning [26].

CNNs are NN architectures inspired by the organization of visual cortex of the brain [27]. They utilize convolutional filters that make use of local connections to share weights between the network's units. The convolutional filters are usually followed by feature subsampling (pooling) operation to conform convolutional layers allowing to extract invariant high-level features from input data (usually, 2D images). The typical CNN architecture consists of one or more convolutional layers (each one, in turn, composed of several convolutional filters and a pooling layer), followed optionally by one or more densely connected layers and ended up with a softmax or sigmoid output layer for supervised learning. Although CNNs have achieved state-of-art performance in computer vision tasks [27, 28], application of CNNs on bioinformatics or, more specifically, genomics data is still an emerging field. Recent works have applied CNN on raw DNA sequences to address diverse predictive problems, such as prediction of specificities of DNA- and RNA-binding protein, prediction of chromatin marks from DNA sequences, or prediction of DNA methylation states [29–31]. CNNs have been also successfully applied to proteomics [32, 33] or multi-omics data [34, 35] in different classification or prediction scenarios.

The application of CNNs to gene-expression data has many limitations, derived from the unstructured nature of these data. In this way, in a recent study [35], multi-layer perceptrons (MLP) and CNNs were comparatively tested—together with linear discriminant analysis (LDA), logistic regression (LR), naive Bayes (NB), random forest (RF) and support vector machines (SVM)—on more than 30 gene-expression datasets to predict disease stages or to discriminate between diseased samples and normal samples. The results in [35] showed that CNNs were among the methods with the worst performance rates for the majority of the datasets analyzed, while MLPs achieved the highest overall accuracy among all methods tested. Although these results could seem surprising, they are somehow obvious, as the effectiveness of CNNs is based on how receptive fields (i.e. convolutional filters) exploit local motifs present in data. In computer vision applications, CNNs deal with images consisting of local regions containing "spatially coherent" pixels, i.e. adjacent pixels in a same image are not independent but they share specific information, so that local patterns can be extracted by the convolutional layers. Consequently, if the pixels in the images were rearranged, so that their relative position changed, it could negatively affect feature extraction and performance of the CNNs [36]. For their part, gene-expression samples lack local motifs since adjacent genes in a same sample are independent, i.e. gene-expression samples do not show "spatial coherence" or, in other words, genes in "local" regions in the sample do not share similar information. This leads us to think that if we rearranged gene-expression data (in an element-wise manner), thus transforming gene-expression samples into better structured patterns—giving them an"image form" by

putting together those genes that share domain-specific information—, CNNs could extract local high-level features and achieve higher performance rates when tackling predictive or classification problems. Following this idea, in [37] Lyu and Haque presented the first preliminary approach to transform gene-expression vectors into two-dimensional images, which were subsequently used to train a CNN architecture for tumor-type classification. They utilized the relative position of the genes in the chromosome, i.e. the transcription locus, as the rearrangement criterion to create "gene-expression images". In [38] Ma and Zhang outline the general ideas of *OmicsMapNet* approach, in which omics data of individual samples are first rearranged into two-dimensional images in which molecular features related to functions, ontologies, or other relationships are organized in spatially adjacent and patterned locations. They apply their proposed methodology to RNA-Seq expression data of TCGA diffuse glioma samples and train a CNN to predict malignancy grade of tumor samples. More recently, Sharma and collaborators have presented *DeepInsight* [36], a general methodology to transform a non-image data into an image for CNN architectures, which the authors apply on different kinds of non-image datasets, namely RNA-seq, vowels, text, and artificial data. Nevertheless, unlike previous works, they do not use domain-specific information to rearrange input feature vectors, but they use instead a much more general pipeline consisting of an initial KPCA (kernel principal component analysis) or t-SNE (t-distributed stochastic neighbor embedding) projection, followed by the application of the convex hull algorithm and an image rotation operation. DeepInsight produces high classification accuracy on a test set of RNA-seq data (obtained from the TCGA public dataset) and, as reported in [36], outperforms other ML methods—i.e. RF, Ada-Boost and decision trees—in a simple tumor-type classification problem of not direct clinical relevance.

On the other hand, training a CNN with gene-expression data, not only implies having to rearrange the data to provide them with structure for convolutional filters to take advantage of local information patterns, but it also involves dealing with the curse of dimensionality inherent to genomics data. Training a CNN architecture—or, more in general, a DL model—with a small set of samples composed of a large number of features (i.e. $N \gg M$) can lead the model to dramatic over-fitting issues. To counteract the enormous imbalance between the number of samples and characteristics, several approaches have been proposed in the literature, which can be summarized into two general categories: data augmentation (DA) and TL strategies. While DA consists in expanding the size of the training set by using different methods and procedures (e.g., introducing affine or perspective transformations for images or using much more complex and general pipelines based on variational auto-encoders [39] or generative adversarial neural networks [40]), TL approaches aim at pre-training the model on a similar *base* dataset to finally fine-tune the model on the *target* dataset. For the TL approach to work properly, the *base* dataset must contain a much greater number of samples than the final *target* dataset. TL has been successfully applied to computer vision, text classification or software error detection, among other domains [41].

Following the preliminary ideas outlined in [38], in this paper we propose a methodology to rearrange RNA-seq data by using a biological criterion aimed at transforming gene-expression vectors into "gene-expression images" in which the relative position of the genes is driven by their molecular function. The KEGG ontology database [42, 43] has been used to query the BRITE hierarchies corresponding to all the genes in RNA-seq tumor samples, which are subsequently rearranged by mapping the tree-shape hierarchies onto a two-dimensional image. A *treemapping* [44] algorithm has been used to display the hierarchical data onto images by using nested rectangles, representing KEGG functional categories and sub-categories, which are ultimately composed of genes. We then follow a TL approach to train a CNN architecture on these "gene-expression images" for cancer survival prediction. Using RNA-seq data

obtained from the The Cancer Genome Atlas (TCGA) program (https://www.cancer.gov/about-nci/organization/ccg/research/structural-genomics/tcga) we pre-train the CNN on a dataset composed of samples of 31 different cancer types, with the exception of lung cancer. Finally, lung cancer RNA-seq samples are used to fine-tune the resulting pre-trained CNN architecture to get high performance rates in a fixed-time survival predictive problem arranged as a binary classification task, thus outperforming MLNNs and other ML approaches analyzed comparatively.

Unlike other predictive problems addressed usually with ML or DL models on the basis of gene-expression data, such as tumor type classification or tumor/non-tumor binary classification, survival prediction deserves special attention for its high clinical relevance, since survival analysis allows physicians to stratify cohorts of patients in order to have more accurate prognosis estimation and establish more precise and effective treatments. In this study, the clinical outcome to be predicted by CNN is *progression free interval* (PFI), which is one of the clinical variables present in Pan-Cancer TCGA dataset, as part of the following phenotypical features related to survival and provided for all the samples in the dataset (with the exception of some missing values): *overall survival* (OS), *disease-specific survival* (DSS), *disease-free interval* (DFI) and PFI. Using OS or DSS demands longer follow-up times, thus in many clinical studies DFI or PFI are used. As concluded by a recent validation study [45], PFI is the best-supported survival variable that can be derived accurately from TCGA available data, as it has been determined to be reliable for 31 out of 33 TCGA cancer types (see Table 3 in [45]). It establishes time (number of days) for patient having new tumor event, whether it was a progression of disease, local recurrence, distant metastasis, new primary tumor, or the patient died with the cancer without new tumor event, including cases with a new tumor event whose type is unknown (see [45] supplementary material).

The rest of the paper is organized as follows. Section *Materials and Methods* describes firstly how the gene-expression datasets used within the analysis are collected and preprocessed; secondly, the methodology proposed here to generate the gene-expression images, by using the KEGG BRITE functional hierarchies as the criterion followed to rearrange gene-expression vectors into images, is explained in details; thirdly, the problem of PFI prediction is presented in this section; then, we introduce the different DL models as well as the TL approach proposed in this study; finally in this section, other traditional ML approaches are presented, as well as the cross-validation (CV) strategy used to compare these models with the DL-with-TL approach. In Section *Results and Discussion* the results obtained by the different approaches are comparatively analyzed and discussed and, finally, some conclusions are provided in the last section of this manuscript.

Finally, for reproducibility purposes, all the data and code needed to replicate our work, is publicly available at https://github.com/guilopgar/GeneExpImgTL.

## Materials and methods

The workflow of our TL approach is shown in Fig 1, and each phase of the pipeline is described in the next subsections.

### Data collection and pre-processing

In this work, the Pan-Cancer dataset from the TCGA project was used [46], accessed from the UCSC Xena data browser (https://xenabrowser.net/datapages/). The Pan-Cancer dataset comprises $\sim 11K$ RNA-Seq gene-expression samples from 33 distinct tumor types, measured by $log_2(TPM + 0.001)$ transformed RSEM values. Every sample contains the expression values of 60498 input variables (transcripts). In order to reduce the unmanageable number of initial

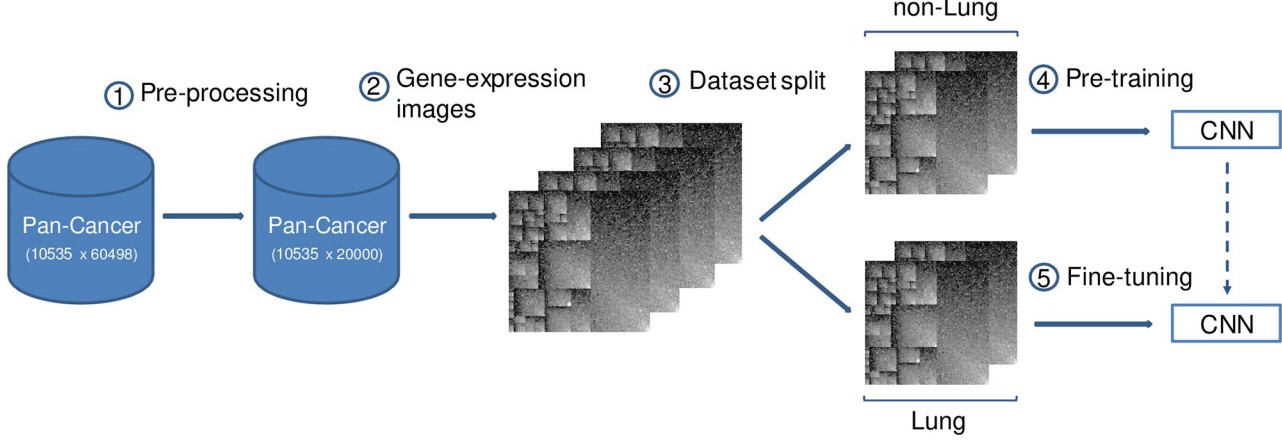

**Fig 1. Workflow of the DL-with-TL approach.**

features, we performed an unsupervised feature selection approach. Using the standard deviation (SD) metric, we firstly removed the uninformative variables with constant expression values across all the samples. Then, according to the median absolute deviation (MAD), only the top $20K$ most variably expressed genes were retained, obtaining a final dataset composed of 10535 samples and $20K$ features.

## Gene-expression images

In this study, our main goal is to transform the gene-expression vectors contained in the Pan-Cancer dataset into images that can be exploited by CNN models. To do so, we rearranged the positions of the genes using a biological criterion. In particular, KEGG BRITE hierarchical information was used to determine the specific location of every gene inside the images.

This second phase of the workflow is divided into two sub-stages: the creation of a tree-shape structure that associates KEGG BRITE functional information to the genes contained in the Pan-Cancer dataset (see section 'KEGG BRITE functional hierarchies'), and the generation of the gene-expression images using the previously created hierarchical structure (see section 'Treemapping procedure').

**KEGG BRITE functional hierarchies.** KEGG is a collection of databases which aims at describing biological entities from a systems-biology perspective. Among them, KEGG BRITE is an ontology database that captures functional hierarchies of KEGG objects, such as molecules, cells, drugs and genes [43]. The objective of this first sub-stage of the second phase of the workflow is to create a tree-shape structure that links the hierarchical functional information contained in KEGG BRITE database to the genes present in the Pan-Cancer dataset.

With the aim of generating this hierarchical structure, we firstly downloaded the KEGG BRITE reference hierarchy file, by making use of the REST-style KEGG API (http://rest.kegg. jp/get/br:br08902). Since we are only interested in the functional hierarchies associated to the human genes contained in the pre-processed Pan-Cancer dataset (see section 'Data collection and pre-processing'), the mapping procedure depicted in Fig 2 was performed. The first step of this process consists in linking human genes with KEGG BRITE functional hierarchies, by using a table that maps the KEGG BRITE ID associated with each functional hierarchy to KEGG human gene IDs. This table was also accessed via the KEGG API (http://rest.kegg.jp/link/hsa/brite). Then, in order to link KEGG human gene IDs with the specific genes present in the Pan-Cancer samples, which are in ENSEMBL notation [47], a double casting operation

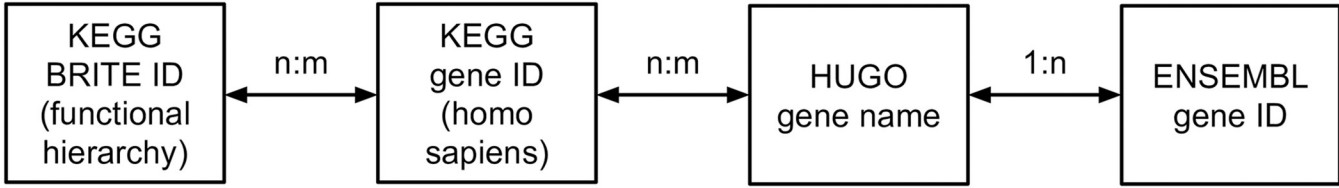

**Fig 2. Mapping procedure.** Mapping between different gene notations to get the functional hierarchies associated to human genes contained in Pan-Cancer dataset (the cardinality of the relation between each two notations is also given).

is needed. On the one hand, a table accessed from the KEGG API (http://rest.kegg.jp/list/hsa) is used to convert the KEGG human gene IDs into the corresponding HUGO gene names [48]. At the same time, ENSEMBL IDs of the Pan-Cancer genes are also converted to HUGO gene names by using a mapping table created by the European Bioinformatics Institute (EBI) (ftp://ftp.ebi.ac.uk/pub/databases/genenames/new/tsv/hgnc_complete_set.txt). In this way, the correspondence between KEGG BRITE functional hierarchies and Pan-Cancer genes can be finally established.

Once the mapping procedure was completed, 45 KEGG BRITE functional hierarchies were associated to 7509 genes from the pre-processed Pan-Cancer dataset. As a result, the tree structure described in Fig 3 was obtained. Using the hierarchical information contained in KEGG BRITE reference hierarchy file, the 45 functional hierarchies were grouped into four major functional categories: "Orthologs and modules", "Protein families: genetic information processing", "Protein families: metabolism" and "Protein families: signaling and cellular processes", which all belong to the "Genes and Proteins" group, i.e. the root of the tree. The leaves of the tree correspond to genes. As it can be observed in Fig 2, a single gene from the Pan-Cancer dataset can be associated with multiple KEGG BRITE functional hierarchies. This is the reason why the number of leaves is 17723, though there are only 7509 distinct genes in the tree.

Finally, from the entire set of $20K$ genes contained in the pre-processed Pan-Cancer dataset, we selected a subset composed of 7509 annotated genes, thus having a final Pan-Cancer dataset of 10535 samples and 7509 features.

**Treemapping procedure.** Fig 4 outlines the second sub-stage of the procedure designed here to transform gene-expression vectors into images. It aims at mapping an unstructured gene-expression vector $\mathbf{g_i} = (g_{i1}, g_{i2}, \ldots, g_{in})$, with $\mathbf{g_i} \in \mathbb{R}^n$, corresponding to a Pan-Cancer sample $i$ with $n$ genes (transcripts), into a structured gene-expression image $\mathbf{G}_i \in \mathbb{R}^{r \times c}$, with $r$ and $c$ being the number of pixels in the rows and columns of $\mathbf{G_i}$, respectively. The locations (in terms of row and column) of every gene-expression value $g_{ij}$ in $\mathbf{G_i}$ are determined by the KEGG BRITE functional hierarchies to which the corresponding gene is associated (see also Fig 3) as well as by the mean expression value of the gene across all Pan-Cancer samples, as described below. For this purpose, the KEGG BRITE functional-hierarchies tree—obtained as a result of the procedure explained in the previous subsection—is firstly converted into a *KEGG BRITE functional-hierarchies image* **T**, with dimensions $r \times c$, by means of the *treemapping* method, explained in the following paragraph. Thus, **T** serves as a sort of "image template" for the subsequent mapping $\mathbf{g_i} \Rightarrow \mathbf{G_i}$. Specifically, in order to generate **T** we used the R package *treemap* (https://cran.r-project.org/package=treemap), which implements the ordered-treemap, pivot-by-size, algorithm [49].

When applied to the KEGG BRITE functional-hierarchies tree, the treemapping procedure used in this study to create **T** works as follows. Given a fixed image size $r \times c$ for **T** and starting from the root of the tree, the treemapping algorithm divides recursively the image into

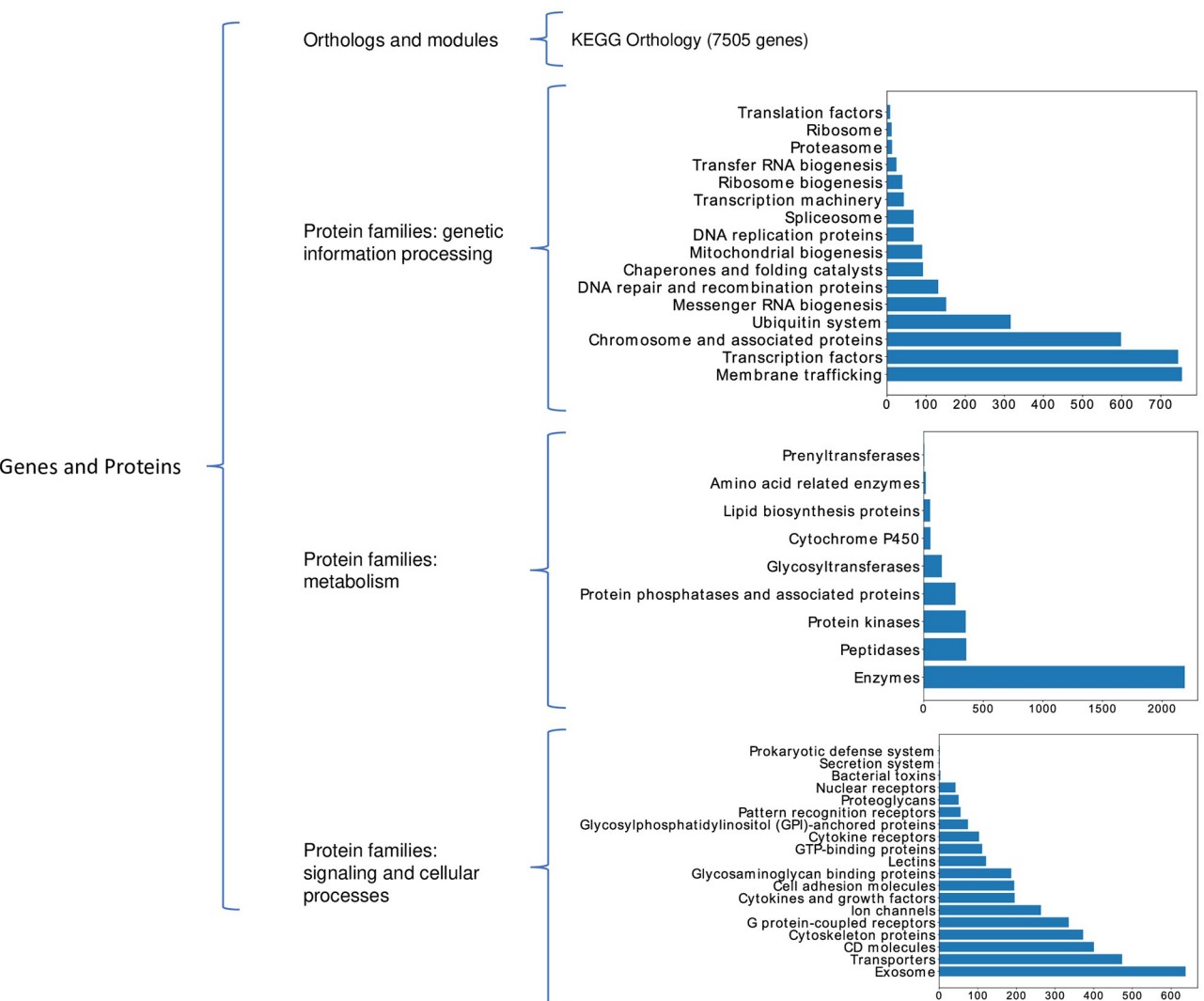

**Fig 3. KEGG BRITE functional-hierarchies tree.** KEGG BRITE functional-hierarchies tree and bar-plots of the degree of each functional hierarchy node, i.e. the number of Pan-Cancer genes associated to each functional hierarchy.

rectangles until all the nodes in the tree are explored, level by level, with each rectangle representing a node in the tree. That is to say that, firstly, the whole image corresponds to the "Genes and Proteins" node, i.e. the root of the tree. The image is subsequently divided into the same number of *functional category rectangles* as the degree of the "Genes and Proteins" node, which is 4 (see Fig 4-*level 1* image and Fig 5). After that, each of the 4 *functional category rectangles* are subsequently subdivided into the same number of *functional hierarchy rectangles* as the degree of the node it represents (see Fig 4-*level 2* image and Fig 6), e.g. the "Protein families: metabolism" category rectangle is subdivided into 9 rectangles, as the degree of the "Protein families: metabolism" node in the tree is 9. Finally, in order to produce the final treemap image **T**, every one of the 45 functional hierarchy rectangles is ultimately subdivided into *gene rectangles* (see Fig 4-*level 3* image), representing the genes, i.e. the leaves of the tree. The relative position of these gene rectangles inside the functional hierarchy rectangles in **T** is determined by their mean expression values across all the samples contained in the Pan-Cancer dataset (see also Fig 7, bottom-right image). Regarding the size of any of the rectangles generated throughout the treemapping procedure, and assuming that all leaves (genes) should

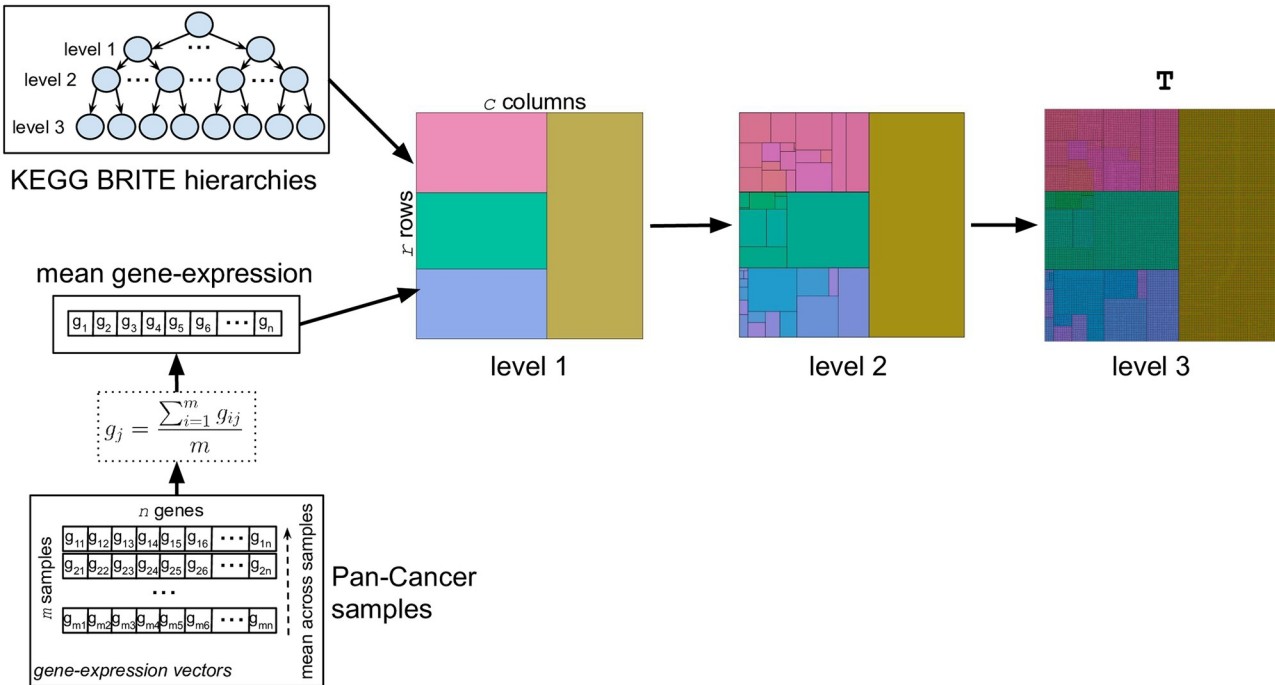

**Fig 4. Treemapping procedure.** KEGG BRITE functional-hierarchies tree and the mean gene-expression values across Pan-Cancer samples are used to get *KEGG BRITE functional-hierarchies image* **T**.

occupy the same rectangular area *a* in the final image, the area *A* of rectangle *rec* is given by $A(rec) = D(rec) \times a$, where $D(rec)$ is the number of descendant leaves of the node in the tree represented by *rec*.

Using the treemapping method described above, the KEGG BRITE functional-hierarchies tree is transformed into the template image **T**, for which the positions of every gene is determined by its locations in the tree as well as its mean expression value across the Pan-Cancer samples. It should be noticed that, since a single gene can have multiple locations in the tree (see section 'KEGG BRITE functional hierarchies'), a unique gene can also occupy multiple positions in **T**.

On the other hand, regarding the dimensions of **T**, an appropriate fixed size $r \times c$ should be selected. If the resolution of **T** is excessively low, the treemapping procedure does not guarantee that all leaves in the tree are contained in the final image. Otherwise, when the resolution of **T** is extremely high, although all leaves—and therefore all genes ($n = 7509$)—are expected to be present in the image, occupying the same rectangular area, the computational cost of processing the subsequently generated gene-expression images is also excessively high. In this work, we have chosen a fixed size of $175 \times 175$ pixels, which was heuristically obtained as the one that allows us to have a final template image **T** containing all 17723 leaves from the functional tree with approximately the same rectangular size, while using a reasonable resolution from a computational perspective.

In Fig 7 bottom-right, a representation of the finally generated image **T** with dimensions $175 \times 175$ is shown. For illustration purposes, the gray colors in this image represent average expression values across all samples from the Pan-Cancer dataset.

Once the KEGG BRITE functional-hierarchies image **T** is obtained, it is used as an image template to create the gene-expression images **G$_i$** corresponding to all samples in the Pan-

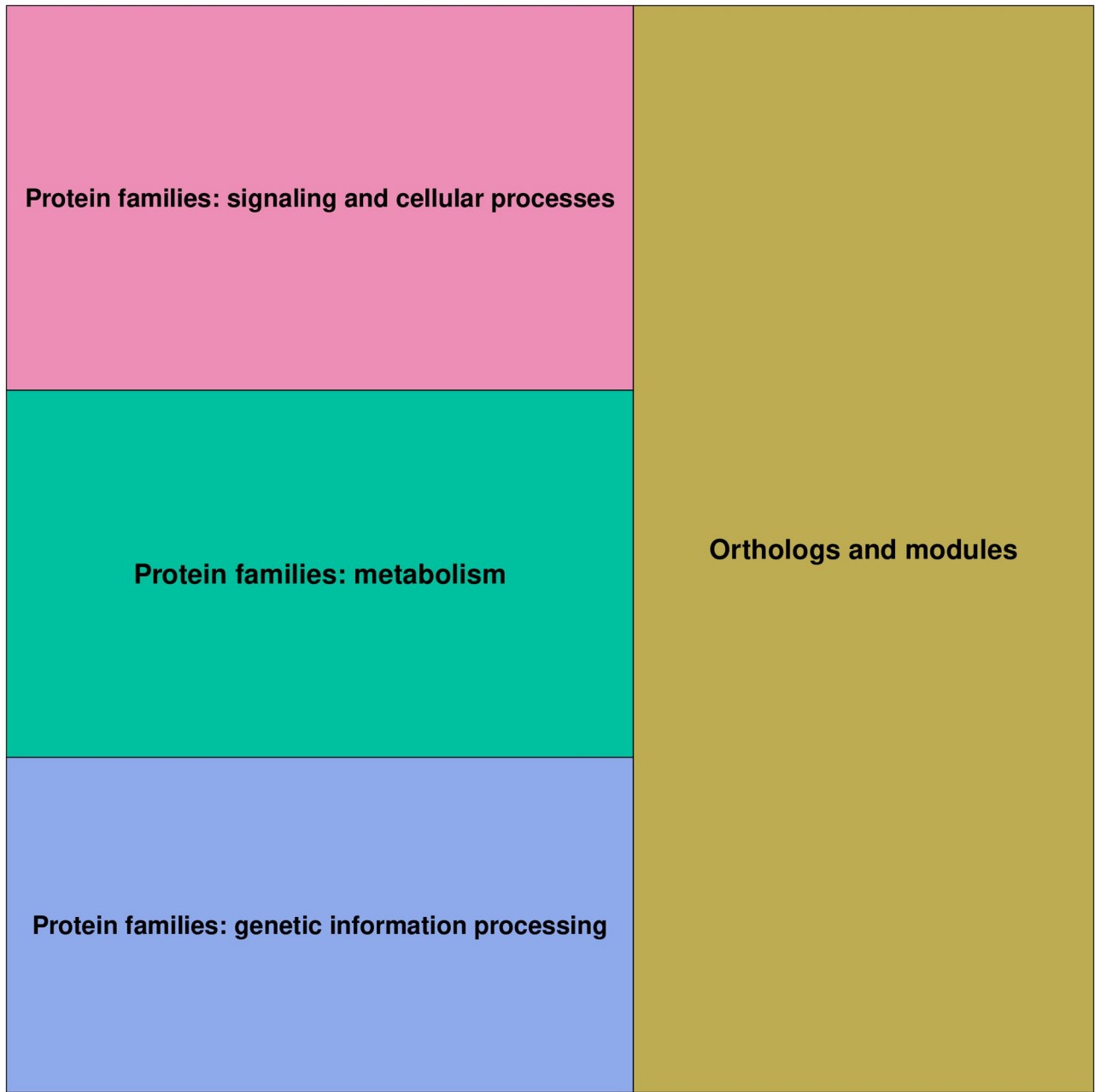

**Fig 5. Level-1 treemap image.** Image generated by subdividing the "Genes and Proteins" initial rectangle into the 4 *functional category rectangles* corresponding to the nodes in the first level of the KEGG BRITE functional-hierarchies tree. (Pseudo-colors are used for representation purposes).

Cancer dataset, in a straightforward manner. For each gene-expression vector (sample $i$) from the Pan-Cancer dataset, we map the expression values to the positions of the genes in $\mathbf{T}$, thus obtaining a unique gene-expression image $\mathbf{G_i} \in \mathbb{R}^{175 \times 175}$ for every sample $i$. In this way, the positions of the genes are the same in all gene-expression images and are determined by $\mathbf{T}$, but the pixel values in $\mathbf{G_i}$ are sample-specific. Images $\mathbf{G_i}$ have all the same resolution as $\mathbf{T}$, i.e. $175 \times 175$ pixels. See Fig 7 (top and bottom-left) for three examples of $\mathbf{G_i}$ images corresponding to three different Pan-Cancer samples.

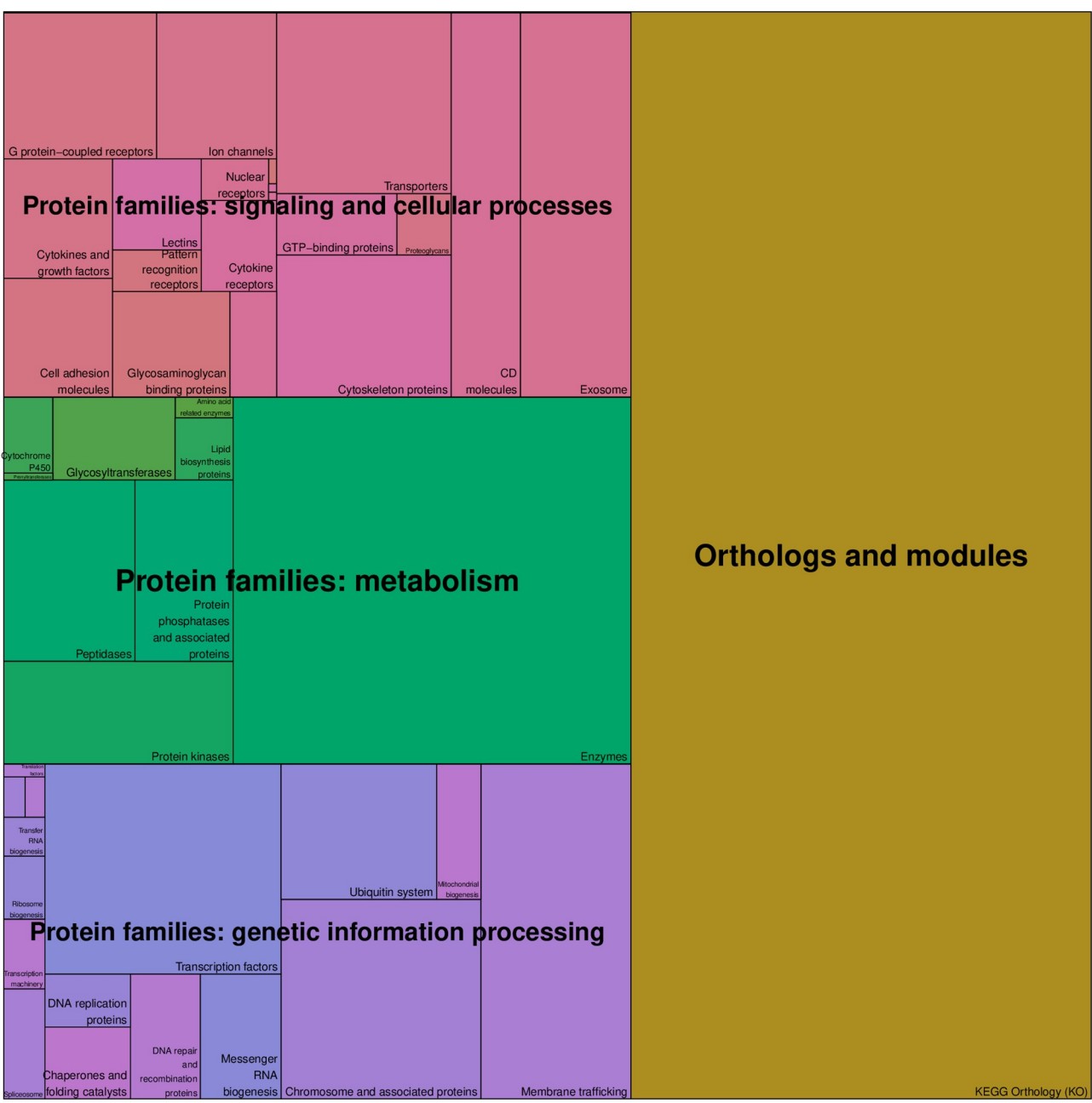

**Fig 6. Level-2 treemap.** Image generated by subdividing the "Orthologs and modules", "Protein families: signaling and cellular processes", "Protein families: metabolism" and "Protein families: genetic information processing" category rectangles into the 45 *functional hierarchy rectangles* corresponding to the second level of the KEGG BRITE tree. (Pseudo-colors are used for representation purposes).

### Lung-cancer PFI prediction

In this study, we train CNN models by means of a TL approach to tackle a specific cancer survival predictive task, namely fixed-time PFI prediction, modeled as a binary classification problem. In this way, given a fixed time point in days $t$, the output of the model represents the probability for a patient to have a new tumor event—whether it was a progression of disease, local recurrence, distant metastasis, new primary tumor, or the patient died with the cancer

TCGA-DK-A6B1-01

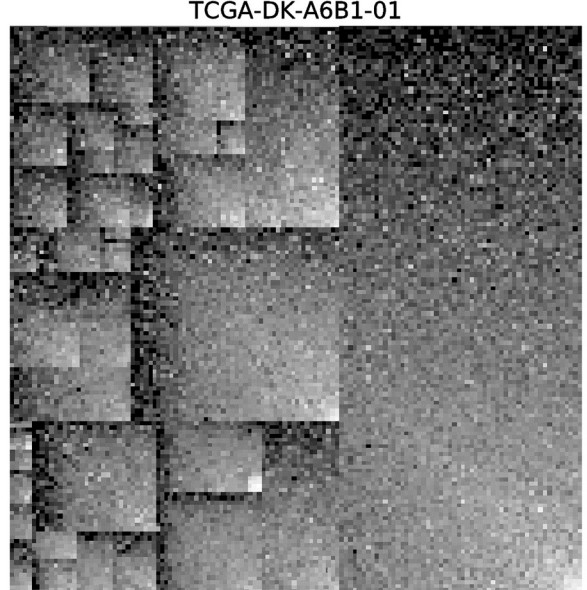

TCGA-IA-A83V-01

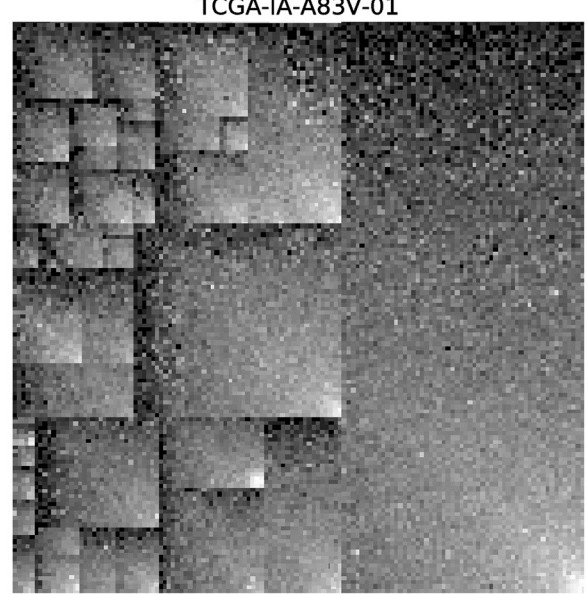

TCGA-YL-A9WJ-01

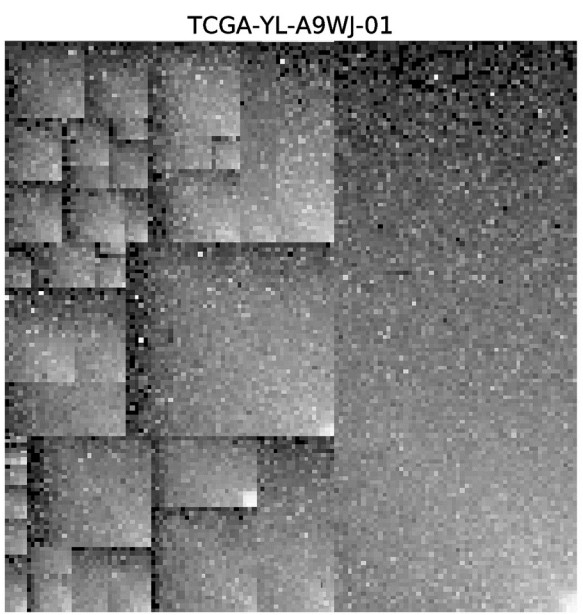

Average expression

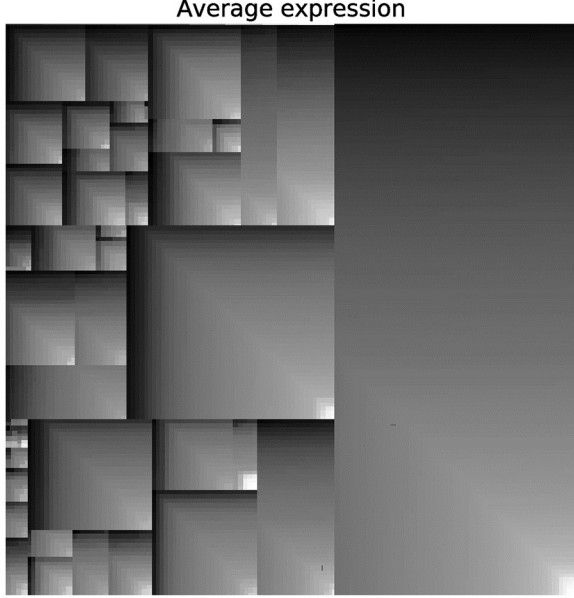

**Fig 7. $G_i$ gene-expression images.** Three examples of $G_i$ gene-expression images (corresponding to Pan-Cancer samples TCGA-DK-A6B1-01, TCGA-IA-A83V-01 and TCGA-YL-A9WJ-01), together with the *average gene-expression image* (bottom right). In the latter image, the color of every gene represents its average expression value across all Pan-Cancer samples, in order to illustrate that, inside each functional hierarchy rectangle, genes are sorted according to their average expression value across all samples. For the four images, gray colors represent expression levels (the lighter the gray level the higher the expression value of the gene it represents).

without new tumor event—during the following *t* days after the initial diagnosis. Particularly, we aim at predicting PFI outcome for lung-cancer samples present in the Pan-Cancer dataset. Since PFI information is available for all Pan-Cancer samples (i.e. not only for lung cancer but also for the remaining cancer types), a TL approach can be adopted. Thus, we split the Pan-Cancer dataset into two distinct subsets: on the one hand, in the *Lung* dataset we include

exclusively the lung-cancer samples—corresponding actually to two different cancer types of Pan-Cancer, namely lung adenocarcinoma (LUAD) and lung squamous-cell carcinoma (LUSC)—, while, on the other hand, in the dataset named *non-Lung* the samples from the remaining 31 Pan-Cancer tumor types are grouped together. The *non-Lung* dataset is used to pre-train the model, initializing the weights of the CNN architecture. Then, the resulting pre-trained model is finally fine-tuned on the *Lung* dataset, thus trying to maximize the predictive capacity of the CNN on the lung cancer patients.

In order to model the available PFI survival information for the Pan-Cancer samples as a binary variable for classification, a suitable value for *t* needs to be chosen. A tumor sample belongs to the positive class if the corresponding patient had a new event before *t* days after the patient was initially diagnosed. On the contrary, if the patient did not have a new tumor event during the previous time period, the corresponding sample belongs to the negative class. Thus, all right-censored samples whose censoring times are below time *t* must be discarded, as it can not be surely determined whether those patients had a new tumor event before time *t*. For that reason, the lower the value of *t*, the higher number of samples that can be retained. However, if the value of *t* is overly low, an extremely imbalanced dataset is obtained, as most of the patients will not have had a new tumor event during such a short time period, leading to a highly underrepresented positive class. In this work, we have estimated a fixed-time point of 230 days ($\sim 7.67$ months), which allows us to retain a high number of samples in both *non-Lung* and *Lung* datasets, while having an assumable class imbalance in both datasets. Hence, we finally obtain a *non-Lung* dataset composed of 7707 samples in which the 12% belong to the positive class, and a *Lung* dataset of 855 samples, with a similar class imbalance, with the 9% of the samples belonging to the positive class.

**DL-with-TL approach.** Following a TL approach, the gene-expression images obtained from the *non-Lung* samples were used to pre-train the CNN model, while the resulting architecture was fine-tuned on the gene-expression images corresponding to the *Lung* samples. Random over-sampling technique was applied both during pre-training and fine-tuning phases of the TL strategy in order to handle the class imbalance of the *non-Lung* and *Lung* cancer datasets. Regarding the model's hyper-parameters optimization, Bayesian optimization [50] along 100 iterations was used to perform automatic tuning of the CNN model's hyper-parameters, such as the number of layers, the dropout rates, the number of filters and kernel size used by every convolutional layer, etc. (see S1 Table for more details). Unlike other TL strategies, for which the hyper-parameters of the pre-trained and the final fine-tuned architectures are independently optimized on the basis of the corresponding *base* and *target* datasets, respectively, in this study all the hyper-parameters of the models were jointly optimized, thus including the pre-training and fine-tuning phase-specific hyper-parameters, such as the learning rates and the batch sizes. To that end, the Bayesian optimization process was uniquely driven by the predictive performance of the final fine-tuned model on the *target* dataset, i.e. the *Lung* dataset, since our main goal is to select the CNN architecture that better predicts the PFI outcome of the lung-cancer samples. To estimate the predictive performance of the CNN, the average Area Under the ROC Curve (AUC) was used as the evaluation metric, by following a 10-repeated 5-fold CV scheme.

Moreover, with the purpose of evaluating the performance of the CNN, which could be derived from its ability of taking advantage of exploiting local patterns present in the $\mathbf{G_i}$ images, we compare it to an alternative DL model that, instead of using the $\mathbf{G_i}$ images as input data, it employs the unstructured gene-expression vectors as the input to the model. With this purpose, a densely-connected multi-layer feed-forward neural network (MLNN) model was also trained for the prediction of lung-cancer PFI. Again, a TL approach was followed by using the *non-Lung* gene-expression vectors to pre-train the MLNN, whereas the gene-expression

vectors from the *Lung* dataset were used to fine-tune the resulting model. A random over-sampling strategy was also applied to overcome the class imbalance of both the *non-Lung* and *Lung* gene-expression datasets. As in the case of CNN, a similar Bayesian optimization procedure with 100 iterations was employed to jointly optimize all the MLNN model's hyper-parameters (see S2 Table for additional information), using average AUC as the evaluation metric—computed across the 50 folds obtained from 10-repeated 5-fold CV scheme on the *Lung* dataset.

**Comparison to ML approaches.** With the aim of assessing the efficacy of the DL-with-TL approach to predict PFI clinical outcome for lung-cancer samples, we compare it to other traditional ML approaches. For ML models to deal with the curse of dimensionality in predictive tasks on gene-expression data—thus avoiding over-fitting issues and increasing overall performance—, feature selection and/or extraction methods have to be included in the workflows. Namely, in this study we analyze the use of three different dimensionality reduction methods—ANOVA feature selection, PCA and KPCA feature extraction—in combination with four traditional ML models—logistic regression (LR), support vector machine (SVM), shallow NN and random forest (RF)—to perform fixed-time PFI binary classification on the lung-cancer Pan-Cancer samples. To overcome the class imbalance of the *Lung* gene-expression dataset, we applied synthetic minority over-sampling technique (SMOTE) [51] immediately after the feature selection/extraction step of the traditional ML workflow. Again, a Bayesian optimization procedure with 100 iterations was used to automatically tune the hyper-parameters of the ML models (see S3 Table for further details), using the mean AUC calculated across the same 50 CV folds of the *Lung* dataset as the model evaluation metric.

Note that, in case of the ML approach, only the raw gene-expression vectors of the *Lung* dataset were used to train the models, since no TL strategy was employed with these models.

## Results and discussion

Given the high class imbalance of the *Lung* dataset (in favor of the negative class), AUC was chosen as the principal metric to evaluate comparatively the performance of the different approaches analyzed here, since it is the preferred metric to deal with very skewed sample distributions, for which we aim at avoiding overfitting to the most frequent class. Besides AUC, five additional performance measures—sensitivity, specificity, F-measure, accuracy and Matthews correlation coefficient (MCC)—were also calculated in order to have a more complete view of the models' behavior. Probability thresholds necessary to compute these metrics for each binary classifier were independently adjusted on each fold during the CV process, by using the ROC-curve to compute the threshold that optimizes the balance between sensitivity and specificity. However, it must be notice that accuracy is not the more appropriate performance measure when dealing with strongly unbalanced datasets, such as the ones we are using for training the models in this study. Thus, in our case the negative class represents more than 90% of the training samples, so that a naive classifier could easily reach more than 90% accuracy (as well as 100% specificity) by classifying all samples as belonging to the negative class.

Since the main objective of this study is to compare the efficacy of different DL and ML approaches when predicting fixed-time PFI for Pan-Cancer lung-cancer samples, in Table 1 mean performance rates on the 50 validation subsets from a 10-repeated 5-fold CV strategy are shown for each approach analyzed here. The first two rows in the table show the results from the DL models—CNN and MLNN—trained by following the TL approach described above, whereas in the last 12 rows the performance rates of the different ML strategies are given. Dashed lines in the table divide the ML models according to the feature selection/extraction method employed by the models (i.e. ANOVA, PCA or KPCA) in order to reduce the number of input features.

**Table 1. Averaged performance measures from 10-repeated 5-fold CV.**

| Approach | Model | AUC | Sensitivity | Specificity | F-measure | Accuracy | MCC |
|---|---|---|---|---|---|---|---|
| DL-with-TL | **CNN** | **0.7326** | **0.6751** | 0.6899 | **0.3101** | 0.6885 | **0.2469** |
|  | **MLNN** | 0.7094 | 0.6117 | **0.7388** | 0.3081 | **0.7269** | 0.2393 |
| ML | **ANOVA-LR** | 0.6958 | 0.6311 | 0.6952 | 0.2859 | 0.6891 | 0.2146 |
|  | **ANOVA-SVM** | 0.7035 | 0.6426 | 0.6968 | 0.2937 | 0.6916 | 0.2266 |
|  | **ANOVA-NN** | 0.6730 | 0.6029 | 0.6840 | 0.2732 | 0.6763 | 0.1990 |
|  | **ANOVA-RF** | 0.6895 | 0.6301 | 0.6760 | 0.2751 | 0.6716 | 0.2012 |
|  | **PCA-LR** | 0.6808 | 0.6196 | 0.6832 | 0.2814 | 0.6772 | 0.2049 |
|  | **PCA-SVM** | 0.6736 | 0.5968 | 0.6923 | 0.2827 | 0.6832 | 0.2064 |
|  | **PCA-NN** | 0.6474 | 0.5441 | 0.7185 | 0.2681 | 0.7019 | 0.1831 |
|  | **PCA-RF** | 0.6498 | 0.5516 | 0.7067 | 0.2654 | 0.6920 | 0.1807 |
|  | **KPCA-LR** | 0.6788 | 0.6658 | 0.6219 | 0.2558 | 0.6260 | 0.1800 |
|  | **KPCA-SVM** | 0.7007 | 0.6589 | 0.6757 | 0.2895 | 0.6742 | 0.2218 |
|  | **KPCA-NN** | 0.6596 | 0.5710 | 0.7005 | 0.2673 | 0.6881 | 0.1885 |
|  | **KPCA-RF** | 0.6650 | 0.6047 | 0.6808 | 0.2728 | 0.6735 | 0.1954 |

According to the average AUC, both CNN and MLNN outperform all ML approaches. In particular, the best AUC (0.7326) is obtained by CNN model. In addition, CNN also obtains the highest value for the sensitivity metric (0.6751), while giving the best balance between sensitivity and specificity (0.6899), which is a remarkable result given the huge class-imbalance present in the *Lung* dataset (with 91% negative samples and only 9% positive samples). Among the ML strategies analyzed, the combination of ANOVA feature selection procedure and SVM classifier produces the best results (0.7035 AUC), followed by combination of KPCA feature extraction method with SVM model (0.7007 AUC). In Fig 8A, a box-plot shows the

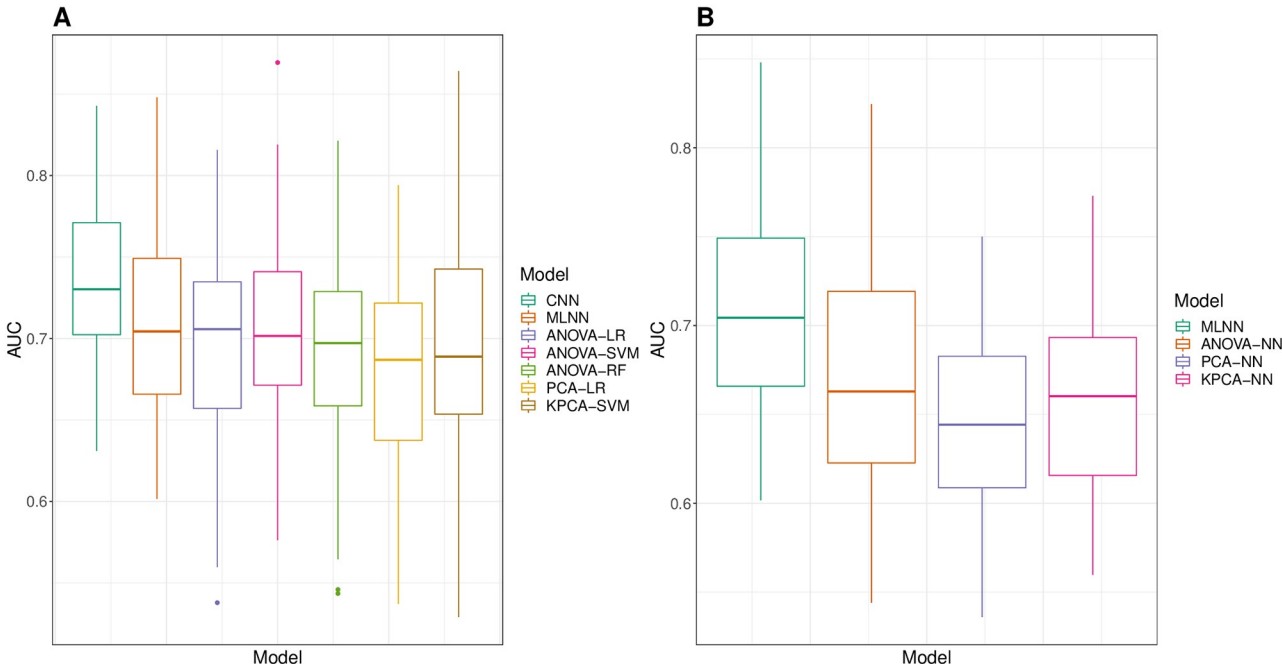

**Fig 8. Box-plots of the distribution of 10-repeated 5-fold CV AUC validation values.** A: Results from the two DL-based approaches (CNN and MLNN with TL) and the five ML approaches that gave the highest average AUC values. B: Results from all the feed-forward NN-based approaches: MLNN with TL vs traditional NN approaches.

distribution of the 10-repeated 5-fold CV AUC validation values obtained by the two DL models as well as the "top five" ML approaches, i.e. those ML strategies that gave the five highest average AUC values.

With the intention of comparing the performance of the CNN model—trained on the gene-expression images $G_i$, generated by means of our proposed workflow (see Section Materials and methods)—with the performance of the rest of the models—trained on raw gene-expression vectors—in a more exhaustive way, a series of statistical tests were carried out. In this way, the non-parametric Wilcoxon signed-rank test was employed to compare the distribution of the 50 validation AUC values obtained by CNN to the distribution of the AUC values obtained by each one of the 13 remaining approaches, thus performing 13 different pair-wise comparisons. Hochberg procedure was applied to the resulting p-values to perform multiple-tests correction [52]. Significant p-values were obtained in all comparisons ($p < 0.001$), hence accepting the alternative hypothesis—i.e. the performance of the CNN model is significantly greater—in all cases. This confirms the superior performance of the CNN (with a TL strategy) over the rest of the approaches analyzed.

We also made an additional series of comparisons in order to evaluate the effectiveness of the TL approach proposed in this work, which can be summarized as leveraging information from a large collection of gene-expression samples belonging to 31 different Pan-Cancer tumor types (i.e. samples in the *non-Lung* dataset), to finally predict fixed-time PFI for a specific tumor type, namely lung cancer (i.e. for samples in the *Lung* dataset). With this purpose, we aimed at comparing the performance of a same ML model, specifically a densely-connected NN, when it is adapted either to a DL-with-TL approach or to a traditional ML strategy. On the one hand, when adapted to a DL-with-TL approach, the NN model takes the form of MLNN and makes use of batch normalization [53], dropout [54] and rectified linear units [55] to take full advantage of the *non-Lung* dataset to pre-train the architecture, as well as the *Lung* samples to finally fine-tune the model. On the other hand, since only the *Lung* dataset is used to train the models in the traditional ML pipelines analyzed in this study, the traditional NN approach has been settled as a shallow feed-forward network preceded by a feature selection/extraction stage, which aims at overcoming over-fitting issues derived from the curse of dimensionality as well as improving predictive performance.

Therefore, in order to assess the efficacy of the TL approach, we compared the distribution of the 50 validation AUC values obtained by MLNN to the distribution of the AUC values obtained by each one of the three ML approaches in which a shallow NN was used as the final classifier: ANOVA-NN, PCA-NN and KPCA-NN strategies. In Fig 8B, a box-plot shows the four corresponding AUC distributions. Considering the average AUC values given in Table 1, MLNN (0.7094) outperforms the rest of the NN-based approaches: ANOVA-NN (0.6730), PCA-NN (0.6474) and KPCA-NN (0.6596) ML models. Again, using the Wilcoxon signed-rank test (with Hochberg correction), significant p-values were obtained in all *MLNN vs x-NN* comparisons, confirming that MLNN performs significantly better than ANOVA-NN ($p < 0.001$), PCA-NN ($p < 0.001$) and KPCA-NN ($p < 0.001$). Finally, it must be also noticed that the highest values for the six performance measures examined in Table 1 were obtained by the two DL models trained by following the TL strategy, which reinforces the potential of the TL approach.

The workflow proposed in this work has been designed to train CNN architectures by using gene-expression images $G_i$ obtained by following a domain-specific criterion for rearranging gene-expression vectors. In combination with a TL approach, our strategy has demonstrated its effectiveness in predicting fixed-time PFI for Pan-Cancer lung-cancer samples. Thus, KEGG BRITE hierarchical information has driven the rearrangement of the positions of the genes in the images, in such a way that a CNN model can subsequently exploit the

local gene-expression motifs present in the $G_i$ images to solve a relevant cancer-survival prediction task, such as fixed-time PFI classification. In addition to KEGG BRITE information, we have also used a criterion based on the average expression value to determine the location of every gene inside the $G_i$ images, since the relative position of the genes inside the KEGG BRITE functional-hierarchy rectangles was determined by their mean expression values across all the Pan-Cancer samples (see Section Materials and methods).

In order to evaluate to what extent using domain-specific information from KEGG BRITE functional hierarchies has contributed to the emergence of structured patterns in the images—from which a CNN can extract high-level features useful for classification—, we performed an additional experiment. In this way, we generated a new set of gene-expression images, that we name $M_i$, for which a much simpler criterion was used to rearrange gene positions inside these new images. Namely, the position of each gene in $M_i$ is solely determined by its mean expression value across all the Pan-Cancer samples, thus genes are sorted in the images (from top-left to down-right corners) in ascending order of mean expression value (see Fig 9 for examples of $M_i$ images). No KEGG BRITE hierarchical information was considered for this new rearrangement criterion. Then, we evaluated the performance of the CNN model when using gene-expression images $M_i$ to predict fixed-time PFI for lung-cancer samples, by following the same TL approach described in Section *DL-with-TL approach*. Using $M_i$ images, the CNN model obtained an average AUC of $0.6998 \pm 0.0584$ (from 10-repeated 5-fold CV). Again, making use of the Wilcoxon signed-rank test (with Hochberg correction), we compared the performance of the CNN model when trained either with $G_i$ or $M_i$ images, resulting in a significantly better performance ($p < 0.001$) of the CNN model that takes advantage of the KEGG BRITE hierarchical information to rearrange the positions of the genes in the input images $G_i$.

Additionally, we also generated a new collection of gene-expression images, called $R_i$, for which we rearranged the genes in a completely random order, so that the resulting images lack any local information that CNNs could exploit to extract high-level characteristics. When trained with $R_i$ gene-expression images, the CNN model obtained a mean CV AUC of $0.6996 \pm 0.0492$, a significantly inferior performance ($p < 0.001$) than the one achieved by the original CNN version trained on the $G_i$ images. In fact, either when using $M_i$ or $R_i$ images, the CNN behaves in the same manner, which denotes that rearranging the positions of the genes by exclusively considering their mean expression values—without any other domain-specific information—does not provide any benefit to the CNN, showing that the network is not capable of extracting high-level characteristics that contribute to improve classification performance. As future work, we could explore whether alternative gene rearrangement strategies based on leveraging other domain-specific information—such as gene chromosome transcription loci, metabolic or signaling pathway information, or even other functional hierarchies schemes—would lead the CNN to similar or better performance results than those obtained with the gene-expression images $G_i$ proposed in the current study.

On the other hand, in this work, we aim at applying a DL model to solve a cancer survival prediction task on a concrete tumor type, thus using a scarce gene-expression dataset. In order to overcome the over-fitting issues derived from the curse of dimensionality, we followed a TL approach. Thus, by pre-training the model on a large collection of $7.7K$ samples —from 31 Pan-Cancer tumor types, different from lung cancer—and fine-tuning the resulting architecture on a reduced dataset of 855 lung-cancer samples, we were able to transfer the high-level feature extractors of the pre-trained general model into a final CNN model which tackles a much more specific task, i.e. lung-cancer survival prediction. In contrast to the results obtained in our previous work [24], this TL strategy has shown to be effective not only for the CNN model but also for a fully-connected feed-forward NN. In this way, our densely-connected NN model (MLNN) has shown to perform significantly better when

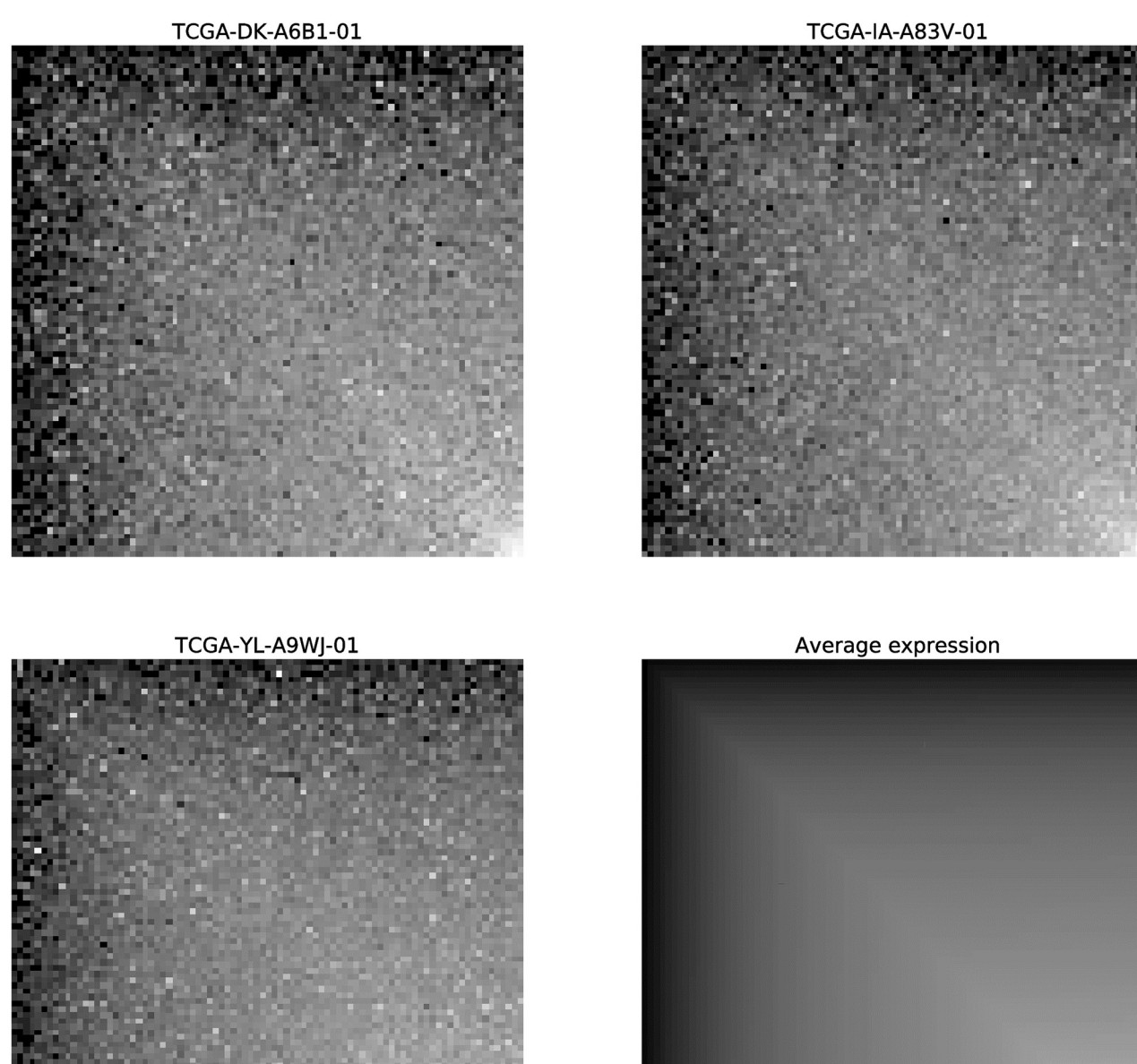

**Fig 9. $M_i$ gene-expression images.** Three examples of $M_i$ gene-expression images (corresponding to Pan-Cancer samples TCGA-DK-A6B1-01, TCGA-IA-A83V-01 and TCGA-YL-A9WJ-01), together with the *average gene-expression image* (bottom right). To generate the images, genes have been sorted by their average expression value across all Pan-Cancer samples. For the four images, gray colors represent expression levels (the lighter the gray level the higher the expression value of the gene it represents).

adapted to our DL-with-TL approach than when utilized within the framework of a traditional ML pipeline that did not make use of any TL strategy. Unlike our previous work [24], in which a similar TL scheme was followed to deal with a more cancer-specific classification task—i.e. prediction of breast tumor intrinsic subtypes—, the TL approach has now been applied inside a more general predictive framework, thus giving better results. Specifically, since PFI information is available for all Pan-Cancer samples, a completely supervised TL

approach has been able to be designed aiming at solving the same cancer prediction task both during supervised pre-training and fine-tuning phases. This strategy has resulted effective for CNN and MLNN to leverage relevant information extracted from a general dataset composed of samples of many different tumor types to improve cancer-specific prediction, i.e. lung-cancer PFI outcome.

Only a few recent studies [37, 38] have already explored the idea of using biological criteria to transform gene-expression vectors into structured images. In these preliminary works, gene-expression images were generated in a single step, by directly mapping gene-expression values to a fixed set of colors, using domain-specific information to determine the position of every gene inside the images. The main drawback of this particular approach is the loss of information caused by using discrete sets of colors, instead of the original continuous expression values, to determine the pixels values of the images. In contrast to this idea, our novel methodology proposes arranging the gene-expression image generation procedure into two consecutive phases. Firstly, a biological functional hierarchy in the form of a tree-shape structure is converted into a functional-hierarchy image **T**, for which the position of every gene is determined by following a specific biological criterion. Then, gene-expression images are created in a straightforward manner, by mapping the expression values to the positions of the genes in **T**, generating a final set of images in which each pixel represents the continuous gene-expression value of the corresponding gene, thus avoiding the loss of information originated from converting continuous gene-expression values into a set of discrete colors.

With the aim of contextualizing our work, although previous studies have applied CNN models to solve cancer predictive tasks on specific tumor types by using gene-expression data [35, 38], none of them has thoroughly addressed the problematic effects derived from the curse of dimensionality, which are inherent to gene-expression datasets. Previous works make use of different CNN models which have been usually trained by using a reduced set of just a few hundreds of samples. However, in this paper we propose a TL approach to deal with the dramatic imbalance existing between the number of available samples and the number of input features contained in gene-expression datasets. In this way, we have used the gene-expression images obtained from a heterogeneous set of thousands of samples from 31 Pan-Cancer tumor types to pre-train a CNN model. Then, the resulting model is fine-tuned on a reduced dataset containing hundreds of lung-tumor labeled gene-expression images. On the other hand and regarding the cancer predictive task tackled in this work, unlike other general (and relatively simple) predictive problems addressed in former studies—such as tumor type multiclass classification or tumor/healthy binary classification—, we face a survival-prediction task of higher clinical relevance, namely fixed-time PFI binary classification. To the best of our knowledge, this is the first work that uses gene-expression data to train a CNN model by following a TL strategy to tackle a cancer-survival predictive task. Additionally, giving the generic nature of the methodology proposed here, our DL-with-TL workflow could be seamlessly adapted to be applied to other types of tumors distinct from lung cancer, using gene-expression images generated by following alternative domain-specific criteria, or tackling other survival predictive tasks, such as OS, DSS or DFI prediction.

## Conclusions

In this paper, we have presented a DL approach to predict lung cancer PFI outcome on gene-expression data. Our proposed workflow combines the effectiveness of a CNN model to extract high-level features from structured input data, with the efficacy of a TL strategy that allows to surpass overfitting issues in scenarios with small training datasets composed of high-dimensional samples, like the ones usually found in cancer predictive tasks with gene-expression

data. Since the original RNA-Seq samples lack local structure necessary for convolutional filters to work properly as feature extractors, we have proposed a methodology aimed to rearrange gene-expression data by transforming lineal expression vectors into gene-expression two-dimensional images, from which a CNN can exploit local motifs contributing to improve lung-cancer survival prediction performance. To that end, by using the well-known *treemapping* method, RNA-seq gene-expression vectors are displayed onto images by following a domain-specific criterion for this rearrangement, which makes use of the KEGG ontology database to query the BRITE hierarchies corresponding to all genes in the dataset, which are subsequently rearranged by mapping the tree-shape BRITE hierarchies onto a two-dimensional image.

In order to do the analysis proposed in this study, the Pan-Cancer dataset has been employed in this work to pre-train CNN architectures on a heterogeneous dataset composed of thousands of gene-expression samples obtained from 31 different cancer types. Once pretrained, the resulting CNN has been fine-tuned by using a reduced dataset composed of hundreds of lung-tumor labeled samples. Finally, with the aim of assessing the effectiveness of our DL approach, we have compared it with the performance obtained by several traditional ML approaches that use different dimensionality reduction methods in combination with traditional ML models to perform fixed-time PFI binary classification on the lung-cancer samples.

According to the average CV AUC, our DL approach outperforms all ML methods analyzed in this work. In particular, the best performance rate was obtained by CNN. Among the ML strategies, the combination of ANOVA feature selection procedure and SVM classifier produced the best results. We also made an additional series of comparisons in order to evaluate the effectiveness of the TL approach, by comparing the performance of a densely-connected feed-forward NN when it was adapted either to a DL-with-TL approach (that we called MLNN) or to a traditional ML strategy. Considering the average AUC values obtained, the MLNN outperformed the rest of the NN-based approaches. However, the MLNN did not reach the efficacy rates given by the CNN model trained on the basis of the gene-expression images, hence demonstrating the higher capability of the latter in leveraging the information extracted from other tumor-type samples to extract high-level features that contribute to improve lung-cancer progression prediction.

In future works, given the promising results obtained in this paper using lung-cancer samples, we will try to extend the DL-with-TL approach to other types of tumors different from lung cancer, such as any of the remaining 31 Pan-Cancer tumor types. For instance, the DL-with-TL approach could be applied to breast cancer (BRCA), so that the DL models would be pre-trained with a large collection of non-BRCA samples and then fine-tuned on a reduced set of BRCA input patterns. Moreover, in addition to fixed-time PFI classification, the DL-with-TL approach could also be easily extended to perform any other cancer prediction task that makes use of the survival outcomes present in Pan-Cancer dataset—OS, DSS, DFI and PFI—, such as continuous time PFI prediction, fixed-time OS classification, discrete time DSS predictive analysis, etc. Special attention will be paid to model interpretability, as though some efforts have already been made in this particular field, most of DL models are still considered as "black-boxes". In areas such as precision medicine and oncology, interpretability is mandatory whether computational algorithms aim to be used as decision-support tools.

## Supporting information

**S1 Table. CNN hyper-parameters optimization.**
(PDF)

**S2 Table. MLNN hyper-parameters optimization.**
(PDF)

**S3 Table. ML models hyper-parameters optimization.**
(PDF)

**S1 File.**
(PDF)

## Author Contributions

**Conceptualization:** Guillermo López-García, José M. Jerez, Francisco J. Veredas.

**Data curation:** Guillermo López-García, Francisco J. Veredas.

**Formal analysis:** Guillermo López-García, Leonardo Franco, Francisco J. Veredas.

**Funding acquisition:** José M. Jerez, Leonardo Franco.

**Methodology:** Guillermo López-García, José M. Jerez, Leonardo Franco, Francisco J. Veredas.

**Project administration:** José M. Jerez, Leonardo Franco, Francisco J. Veredas.

**Resources:** José M. Jerez, Leonardo Franco.

**Software:** Guillermo López-García, Francisco J. Veredas.

**Supervision:** José M. Jerez, Leonardo Franco, Francisco J. Veredas.

**Validation:** Guillermo López-García, Leonardo Franco, Francisco J. Veredas.

**Visualization:** Guillermo López-García, Francisco J. Veredas.

**Writing – original draft:** Guillermo López-García, Francisco J. Veredas.

**Writing – review & editing:** Guillermo López-García, José M. Jerez, Leonardo Franco, Francisco J. Veredas.

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
