## [Decision Letter · Decision Letter 0]

16 Jan 2020

PONE-D-19-33176

Transfer learning with convolutional neural networks for cancer survival prediction using gene-expression data

PLOS ONE

Dear Mr Lopez-Garcia,

Thank you for submitting your manuscript to PLOS ONE. After careful consideration, we feel that it has merit but does not fully meet PLOS ONE’s publication criteria as it currently stands. Therefore, we invite you to submit a revised version of the manuscript that addresses the points raised during the review process.

The reviewers argue about the necessity to address the class imbalance problem, the need for a comparison with conceptually similar works, the desire for more information on the feature selection component, a larger description of the technical aspects of algorithm implementation, a clearer demonstration of the originality and impact of the idea.

We would appreciate receiving your revised manuscript by Feb 29 2020 11:59PM. To enhance the reproducibility of your results, we recommend that if applicable you deposit your laboratory protocols in protocols.io, where a protocol can be assigned its own identifier (DOI) such that it can be cited independently in the future. For instructions see: http://journals.plos.org/plosone/s/submission-guidelines#loc-laboratory-protocols

We look forward to receiving your revised manuscript.

Kind regards,

Ruxandra Stoean

Academic Editor

PLOS ONE

Journal Requirements:

Reviewers' comments:

Reviewer's Responses to Questions

**Comments to the Author**

1. Is the manuscript technically sound, and do the data support the conclusions?

Reviewer #1: Yes

Reviewer #2: Yes

2. Has the statistical analysis been performed appropriately and rigorously? 

Reviewer #1: Yes

Reviewer #2: Yes

3. Have the authors made all data underlying the findings in their manuscript fully available?

Reviewer #1: Yes

Reviewer #2: Yes

4. Is the manuscript presented in an intelligible fashion and written in standard English?

Reviewer #1: Yes

Reviewer #2: Yes

5. Review Comments to the Author

Reviewer #1: This is a very interesting work revealing the use of RNA-Seq data for cancer survival prediction in terms of deep learning and conventional ML techniques. I have no doubts about the work done and the methodology that is proposed in terms of transfer learning and survival prediction. Some concerns that need to be addressed before publication by the authors:

The class imbalance problem should be handled in different ways for obtained unbiased results. Under sampling, over sampling and ensemble techniques do exist for performing this task.

The figures need to be updated in terms of resolution, they are not in the preferred format I think.

A comparison table with similar studies-methodologies should be presented in order to clearly highlight the impact of the current methodology and its usage in another studies.

Reviewer #2: The article seems to be sufficiently novel and interesting. The main goal is to propose and apply the DL-with-TL methodology to rearrange RNA sequence data by transforming them into gene-expression images. Based on these images CNN can extract high-level features. Additionally, authors investigate whether extracted knowledge contributes to the extraction of features that improve lung cancer prediction. The results are compared with other selected ML approaches. This approach imply promising results about successful prediction of investigated problem. I think that this paper adhere to the journal standards.

Presented paper is well organised. Abstract of the article is relevant to its content. It is formulated correctly and relates directly to the substantive content of the article. The article is divided into appropriate sections like introduction with some theoretical background, some historical steps, presentation of investigated approach, materials and methods, gathered research results and summary. It also contains a detailed and thematically relevant bibliography.

The technical content of the article is generally understandable. Authors can add information about the result of the selection of features using applied FS methods. How many original features were considered significant. Maybe also describing the technical aspects of the algorithm implementation might be of interest to readers. Methods must be described in sufficient detail for another researcher to reproduce the experiments described. Authors should provide adequate justification that the methods presented in the paper are significantly different from previously presented.

6. PLOS authors have the option to publish the peer review history of their article (what does this mean?). If published, this will include your full peer review and any attached files.

Reviewer #1: No

Reviewer #2: No

---

## [Author Response · Author response to Decision Letter 0]

27 Feb 2020

As indicated by the academic editor, a rebuttal letter responding to each point raised by the academic editor and reviewers has been uploaded as separate file and labeled 'Response to Reviewers'. The content of that file will be copied and pasted here:

PONE-D-19-33176

Transfer learning with convolutional neural networks for cancer survival prediction using gene-expression data

PLOS ONE

We would like to thank the editor and the reviewers for their fair comments and suggestions and for giving us the opportunity to improve and to re-submit our paper. In this revised version of our paper we have addressed all the comments pointed out by the reviewers and emphasised the novel contributions beyond our previous related work.

COMMENTS FOR REVIEWER #1:

We would like to thank this reviewer for her/his comments and suggestions.

The class imbalance problem should be handled in different ways for obtained unbiased results. Under sampling, over sampling and ensemble techniques do exist for performing this task.

As it has been fairly suggested by this reviewer, we conducted additional experiments in order to handle the severe class imbalance of the gene-expression datasets. Different techniques―random under-sampling, SMOTE and random over-sampling―were employed to re-sample the data before feeding it to the predictive models. We analyzed the performance of the models using each technique, and finally selected the one that maximized the mean CV AUC metric. In this way, random over-sampling procedure applied both during pre-training and fine-tuning phases of the TL approach gave the best results both using CNN and MLNN models. When applied in combination with the traditional ML approach, SMOTE gave the highest AUC values. The revised version of the paper has been updated with the new results obtained using random-oversampling and SMOTE in combination with DL-with-TL approach and the traditional ML approach, respectively. Figure 8 has also been updated.

The figures need to be updated in terms of resolution, they are not in the preferred format I think.

Following the submission guidelines, all figures were uploaded as separate files. Besides, we used PACE software, as suggested in Figures guidelines, to ensure all figures met PLOS requirements.

A comparison table with similar studies-methodologies should be presented in order to clearly highlight the impact of the current methodology and its usage in another studies.

To the best of our knowledge, we are the first authors to explore a TL approach using CNN models on gene-expression data to tackle a cancer-survival predictive task. For this reason, the results obtained in this paper are not directly comparable with the results achieved in previous works, as different methodologies are applied as well as completely distinct cancer prediction tasks are tackled. However, we totally agree with this reviewer on the necessity of performing a thorough comparison between our proposed methodology and similar-studies methodologies to lucidly emphasize the impact of our contribution. In this way, the final paragraphs of the Results and Discussion section in the revised version of the manuscript addresses this concern.

COMMENTS FOR REVIEWER #2:

We would like to thank this reviewer for her/his comments and suggestions.

Authors can add information about the result of the selection of features using applied FS methods. How many original features were considered significant.

We considered the number of input features to be selected by the FS method as a hyper-parameter of the traditional ML approach. The range of possible values of this hyper-parameter, as well as the possible numbers of features to be extracted by the feature extraction methods, are described in detail in S3 Table.

Maybe also describing the technical aspects of the algorithm implementation might be of interest to readers. Methods must be described in sufficient detail for another researcher to reproduce the experiments described.

As stated in the revised version of the paper, for reproducing the results obtained in this work, all the data, code and technical descriptions of the algorithm implementation have been made publicly available at https://github.com/guilopgar/GeneExpImgTL

In addition, the Supporting Information file contains a detailed description of the hyper-parameter optimization procedure employed to perform automatic tuning of the hyper-parameters of all the models analyzed in this work.

Authors should provide adequate justification that the methods presented in the paper are significantly different from previously presented.

As it has been fairly suggested by this reviewer, the Results and Discussion section from the revised version of the manuscript contains a detailed description of the novelties and methodological contributions of the current study, as well as a thorough comparison between our proposed methodology and other similar-studies methodologies.

---

## [Editor Report · Decision Letter 1]

3 Mar 2020

Transfer learning with convolutional neural networks for cancer survival prediction using gene-expression data

PONE-D-19-33176R1

Dear Dr. Lopez-Garcia,

We are pleased to inform you that your manuscript has been judged scientifically suitable for publication and will be formally accepted for publication once it complies with all outstanding technical requirements.

With kind regards,

Ruxandra Stoean

Academic Editor

PLOS ONE
---

## [Editor Report · Acceptance letter]

6 Mar 2020

PONE-D-19-33176R1 

Transfer learning with convolutional neural networks for cancer survival prediction using gene-expression data 

Dear Dr. Lopez-Garcia:

I am pleased to inform you that your manuscript has been deemed suitable for publication in PLOS ONE. Congratulations! Your manuscript is now with our production department. 

With kind regards,

on behalf of

Dr. Ruxandra Stoean 

Academic Editor

PLOS ONE